# Functional Properties of Rye Prolamin (Secalin) and Their Improvement by Protein Lipophilization through Capric Acid Covalent Binding

**DOI:** 10.3390/foods10030515

**Published:** 2021-03-01

**Authors:** Zeinab Qazanfarzadeh, Mahdi Kadivar, Hajar Shekarchizadeh, Raffaele Porta

**Affiliations:** 1Department of Food Science and Technology, College of Agriculture, Isfahan University of Technology, Isfahan 84156-83111, Iran; z.qazanfarzadeh@yahoo.com (Z.Q.); kadivar@cc.iut.ac.ir (M.K.); shekarchizadeh@cc.iut.ac.ir (H.S.); 2Department of Chemical Sciences, University of Naples “Federico II”, Complesso Universitario di Monte Sant’Angelo, 80126 Naples, Italy

**Keywords:** secalin, rye prolamin, protein acylation, capric acid, emulsifying agent, foaming agent

## Abstract

Secalin (SCL), the prolamin fraction of rye protein, was chemically lipophilized using acylation reaction by treatment with different amounts of capric acid chloride (0, 2, 4, and 6 mmol/g) to enhance its functional properties. It was shown that SCL lipophilization increased the surface hydrophobicity and the hydrophobic interactions, leading to a reduction in protein solubility and water absorption capacity and to a greater oil absorption. In addition, SCL both emulsifying capacity and stability were improved when the protein was treated with low amount of capric acid chloride. Finally, the foaming capacity of SCL markedly increased after its treatment with increasing concentrations of the acylating agent, even though the foam of the modified protein was found to be more stable at the lower level of protein acylation. Technological application of lipophilized SCL as a protein additive in food preparations is suggested.

## 1. Introduction

Wheat (*Triticum aestivum* L.) and rye (*Secale cereale* L.), the most commonly used grains in bread production [1], are closely related in taxonomy and, accordingly, their kernel endosperms contain homologous storage proteins [2,3]. However, prolamin occurring in rye grains, and alcohol-soluble protein fraction called secalin (SCL), has not been so widely studied up to now, in relation to its functionality in food systems, as wheat prolamin has been. According to previous studies, the electrophoretic pattern of SCL shows four groups of polypeptides, indicated as high molecular weight (HMW, >100 KDa), γ-75 KDa, ω (50 KDa), and γ-40 KDa proteins, respectively [4]. Moreover, quantitative amino acid analysis evinced that glutamic acid and proline are the predominant components in SCL, followed by leucine, phenylalanine, and serine [4]. Because of the SCL amphiphilic behavior due to the presence of wide regions of protein β-sheets, ethanol (70% *v*/*v*) is the most common solvent used for its extraction. In addition, ethanol leads to the unfolding of the α-helical structures increasing the extent of the β sheets and, consequently, exposing a greater number of hydrophilic groups of the protein. It is worthy to note that the β-sheet structural feature could influence SCL functional properties, i.e., those conferred by the protein to a food product [2,3,4,5,6] as the protein hydrophobic behavior and hence its interactions with the aqueous phase are important factors for the protein functionality. Furthermore, it is well known that protein secondary and tertiary structure may change following a chemical modification able to cause alterations in the surface exposure of their amino acids [7]. Therefore, an induced increase in protein hydrophobicity can enable its integration into lipid systems and, consequently, trigger new, as well as ameliorated or impaired, functional properties [6,8].

Protein lipophilization reaction is defined as a chemical or enzymatic structural change of a polypeptide chain by the addition of lipid components, carried out with several different methods and resulting in biomacromolecules with an increased affinity towards non-polar compounds of low or high molecular mass. Acylation by N-hydroxysuccinimide ester, acetic, succinic, or citraconic anhydrides [9,10,11,12,13], fatty acid chlorides [6] or reductive alkylation [14], as well as by attaching hydrophobic amino acids or alcohols [15,16,17,18], are the most frequently used experimental procedures to obtain protein lipophilization. In particular, the chemical acylation of a protein is a quite considerable structural modification that can dramatically influence the ability of the protein to interact with other molecules and, consequently, modify its activity.

The main purpose of this study was to synthesize SCL derivatives by chemical acylation with capric acid chloride (CAC), an organic compound known as a medium-chain fatty acid, and to investigate the consequent changes in SCL solubility, hydrophobicity, water, and fat binding capacity, as well as in its emulsifying and foaming properties.

## 2. Materials and Methods

### 2.1. Materials

SCL was extracted from rye flour (1:5, *w*/*v*) by 70% (*w*/*w*) ethanol containing 0.5% (*w*/*w*) sodium metabisulfite at 50 °C for 1 h and the presence of its four main polypeptide fractions (HMW, γ-75 KDa, ω, and γ-45 KDa) was then confirmed by sodium dodecyl sulfate-polyacrylamide gel electrophoresis (SDS-PAGE) under reducing conditions [4]. The composition of the SCL preparation was the following: ~91% protein (dw, N factor 5.7), ~5.2% carbohydrate (dw), ~5% moisture, ~1% lipid (dw), ~0.3% ash (dw). Corn oil was purchased from a local supermarket, whereas CAC (98% purity) and the fluorescent probe 8-anilino-1-naphthalene sulfonic acid (ANS) were purchased from Merck KGaA Co. (Darmstadt, Germany). All other chemicals and reagents utilized in this study were of analytical grade.

### 2.2. SCL Chemical Lipophilization

Chemical lipophilization was carried out following Schotten–Baumann’s reaction. SCL was dispersed in 0.5 N NaOH (10%, *w*/*v*) and treated at both 25 and 40 °C for 15 min under stirring. Then, the pH was adjusted to 9.0 with 6 N HCl, CAC (2, 4, and 6 mmol/g of SCL) was added dropwise from a stock solution (4.7 M) to the reaction mixture under stirring, and the pH of the mixture was maintained at a value between 8.5 and 9.0 (using a pH meter Model 3310, Jenway, Staffordshire, UK) by 2 N NaOH. The reaction was considered to be completed when no more change in pH was observed. The modified SCL was then precipitated by pH adjustment to 5 using 6 N HCl, and the pellet obtained by centrifugation at 5000× *g* for 10 min was washed and centrifuged again to eliminate the excess of salts. The precipitate was finally freeze-dried, and the unreacted fatty acid present in the lipophilized SCL was eliminated by hexane extraction. 

The preparation of acylated SCL tested to analyze the functional properties of the lipophilized protein was the same previously used to obtain biodegradable films [19]. The acylation efficiency and the protein recovery rate were calculated by measuring the weight and the total nitrogen content of both unmodified and lipophilized proteins [19]. Fourier Transform Infrared (FTIR) spectroscopy analysis (Model Tensor 27, Bruker Optik GmbH Co., Ettlingen, Germany) was also performed to confirm the interaction between SCL and CAC [19]. FTIR spectra of the samples were recorded within the wavenumber range of 400–4000 cm^−1^ at a resolution of 4 cm^−1^ and 32 spectra/scan at a scan speed of 2 mm/s. Lipophilization degree was determined by the elementary analysis using an Elementar Vario EL system (Model Vario EL, Elementar Analysensysteme GmbH Co., Langenselbold, Germany) and calculated on the base of the determined carbon amount. In fact, the increased carbon value compared to the control sample was converted to the interacted CAC amount value or lipophilization degree.

### 2.3. SCL Functional Properties

#### 2.3.1. Surface Hydrophobicity

The surface hydrophobicity index (SHI) of both unmodified and lipophilized SCL was determined using ANS as previously described by Wan et al. [20]. Protein was dispersed (1 mg/mL) in 10 mM phosphate buffer, pH 7.0, and then diluted with the same buffer to obtain different samples with protein concentrations ranging from 0.1 to 0.25 mg/mL. Finally, 4 mL of each diluted sample were mixed with 20 µL of 10 mM phosphate buffer, pH 7.0, containing 8 mM ANS and the fluorescence intensity was measured at λ excitation = 390 nm and λ emission = 470 nm. The SHI was calculated by measuring the slope of the fluorescence intensity vs. the protein concentration.

#### 2.3.2. Solubility Index

Protein solubility index (SI) was determined by a previously described method with slight modifications [21]. Different samples were prepared by suspending both unmodified and lipophilized SCL in distilled water (1.0% *w*/*v*) brought at different pH values (3–11) by 0.5 N HCl or NaOH. The suspensions were stirred for 30 min and their pH was checked again and readjusted to the desired value. The protein content of the supernatants obtained after centrifugation at 10,000× *g* for 20 min was determined by the Bradford assay method. In addition, protein solubility was also analyzed at pH 7 in the presence of two different NaCl concentrations (0.35 and 0.7 M). The SI was calculated by the following equation:
SI (%) = (soluble protein × 100)/initial protein(1)

#### 2.3.3. Emulsifying Properties

Emulsifying activity index (EAI) and emulsion stability index (ESI) were determined according to the method of Pearce and Kinsella [22] with the following modifications: 7.5 mL of either unmodified or lipophilized SCL in 0.1 M phosphate buffer, pH 7, (1.0% *w*/*v*) and 2.5 mL of corn oil were homogenized for 1 min at 14,000 rpm using an IKA ultra-turrax (Model 3725001, IKA^®^-Werke GmbH & Co., Deutschland, Germany) and, then, 50 µL of the prepared emulsion was dispersed into 10 mL of phosphate buffer containing 0.1% SDS (*w*/*v*). The absorbance at 500 nm of the diluted sample was measured at time 0 (A_0_) and after 10 min (A_10_). The following equations were used to calculate EAI and ESI:
EAI (m^2^/g) = (2 × 2.303 × A_0_ × D)/(C × ∅ × L × 10000)(2)
ESI (min) = (A_0_ × 10)/(A_0_ − A_10_)(3)
where D is the dilution factor (200), C is the initial concentration of protein (g/mL), ∅ is the volume fraction of the oil and L is the cuvette path length (m).

#### 2.3.4. Foaming Properties

Foaming capacity (FC) and foam stability (FS) were determined according to the method of Lawhon et al. [23]. Either unmodified or lipophilized SCL samples (1.0% *w*/*v*) were prepared in 0.1 M phosphate buffer at pH 7, homogenized for 1 min at 14,000 rpm using an IKA ultra-turrax and, then, immediately transferred into a 100 mL graduated cylinder. The total volume of foam layers was recorded at 0, 20, 40, and 60 min storage at room temperature. The following equations were used to calculate FC and FS:
FC% = (foam V after homogenization × 100)/initial V of sample before homogenization(4)
FS% = [foam V after storage (at 20, 40, or 60 min) × 100]/initial foam V(5)

#### 2.3.5. Water and Oil Absorption Capacity

Oil absorption capacity (OAC) and water absorption capacity (WAC) were determined according to the method described by Beuchat [24]. Either unmodified or lipophilized SCL samples (500 mg) were dispersed in 5 mL of either distilled water or corn oil and, after stirring by vortex, were left to stand for 30 min. The samples were then centrifuged at 3000× *g* for 10 min and the volume of the supernatants were finally measured. OAC and WAC were expressed as mL of oil or water, respectively absorbed, per g of protein.

### 2.4. Statistical Analysis

Experiments were always carried out in triplicate. Data were processed by Excel 2010 and analysis of variance (ANOVA) was done using the Statistical Package for the Social Sciences (Version 19, SPSS Inc., Chicago, IL, USA) software to determine the significant difference between treatments. Tukey test was used to compare the mean at a 95% confidence level.

## 3. Results and Discussion

Figure 1 shows that the extracted SCL exhibited the well-known protein electrophoretic profile containing four fractions (HMW, γ-75 KDa, ω, and γ-45 KDa) [4].

It is well known that all nucleophilic groups of protein amino acid residues, such as amino, hydroxyl, sulfhydryl, and phenol groups, can be acylated by fatty acid chlorides [25]. Therefore, the lateral chains of the endoprotein amino acids Lys, His, Ser, Thr, Cys, and Tyr could participate in the protein lipophilization reactions forming new O-ester, S-ester, or amide bonds. According to the previous study on the composition and amount (%) of SCL amino acids [4], the rye prolamin was shown to contain about 18% mol Lys/His/Ser/Thr/Cys/Tyr. Although some of the available hydroxyl and amino groups of these amino acids might theoretically be buried inside the protein due to its conformation and, thus, might not be able to participate in the acylation reaction, it was recently reported the SCL ability to be acylated by CAC [19]. This organic compound belongs to the class of the medium-length chain fatty acids and, thus, it is potentially able to react with polypeptides more effectively than longer fatty acids such as myristic or palmitic acids. Moreover, it has been shown that the amount of the acylated SCL increased by increasing CAC content in the reaction mixture [19]. Lipophilization degree, calculated by elementary analysis of the carbon amount occurring in the modified SCL, reached the value of ~4.6 mmol CAC/g of protein by using 6 mmol/g of CAC in the reaction mixture and, at this concentration of the acylating agent, the protein recovery rate and the acylation efficiency were ~91 and ~85%, respectively [19]. These results were also confirmed by FTIR analysis showing the formation of additional ester and amide groups in the SCL samples previously treated with CAC [19] and were consistent with the data reported by Shi et al. [26] on the lipophilization of zein with lauroyl chloride. Therefore, we were stimulated to investigate the effect of SCL treatment with different amounts of CAC (0, 2, 4, and 6 mmol/g protein) on the protein functional properties. 

Since protein surface hydrophobicity is an important factor affecting its functional properties [27], the ability of both unmodified and lipophilized SCL to bind the ANS fluorescent probe was preliminarily investigated. Figure 2 shows that SCL surface hydrophobicity markedly increased as a consequence of protein lipophilization, probably because an increased number of sites accessible to ANS became available following the covalent CAC binding to the protein. The increasing of protein hydrophobicity by acylation reaction has also been reported both by Shilpashree et al. [28] and Schwenke et al. [29] following succinylation of caseins and globulins, respectively.

The solubility of food proteins is an important property for their application in food processing because it can affect both emulsifying and foaming properties [30]. Figure 3A shows the SI of SCL, both unmodified and lipophilized by different CAC amounts, at different pH values. 

The results revealed that the unmodified SCL had the lowest solubility at pH 6 and that the protein solubility increased at pH values lower and higher than pH 6, at which SCL exhibited charges of higher intensity, positive and negative, respectively. A similar solubility profile was observed analyzing lipophilized SCL, even though the incorporation of CAC led to a significant reduction in protein solubility, particularly evident at acidic pH values. Such observed SI decrease could be primarily due to the increased hydrophobic features of acylated SCL related to the covalent incorporation of numerous fatty acid chains, determining a reduction in charged chemical groups and the consequent possible formation of additional hydrogen bonds. Similar results have been reported by Mendoza-Sanchez et al. [8] for bovine α-lactalbumin acylation. However, the highest SI was observed at alkaline pH values, both for unmodified and lipophilized SCL. Therefore, it is conceivable to assume that the lipophilization shifted the isoelectric point of SCL to a lower pH value, most probably because of the reduction of free amino groups after reaction with the fatty acid. Similar effects of acylation were reported in mung bean [31] and soy proteins [32]. Figure 3B shows the effect of high NaCl concentrations on SCL solubility. The SI of unmodified protein was observed to significantly increase when the saline concentration increased up to 0.35 M, probably as a consequence of a salting-in effect due to the small increase of NaCl. In fact, further enhancement of salt concentration resulted in an evident decrease in SCL solubility related to a salting-out opposite effect. Moreover, the interaction of negatively charged chloride ions with positively charged protein molecules led to a decrease in the electrostatic repulsion, enhancing the hydrophobic interactions and the consequent protein aggregation [33]. Conversely, the solubility of the lipophilized SCL seems to be not affected by the presence of high NaCl concentrations, most probably because the surface charges of SCL significantly changed with lipophilization and, consequently, NaCl was unable to influence the solubility of the modified protein. 

The effects of SCL lipophilization on its WAC and OAC, as well as on the EAI and ESI, are reported in Table 1. WAC decreased in parallel with the lipophilization increase determined by the enhancing of CAC amounts in the SCL acylation reaction mixture. The observed WAC decrease is most probably due to the enlargement of the protein hydrophobic regions causing a reduced SCL ability to bind and retain water. Conversely, protein OAC markedly increased after SCL lipophilization. Furthermore, SCL EAI was found to significantly increase when 2 and 4 mmol/g of CAC were present in the acylation reaction mixture, while it significantly decreased at CAC concentrations of 6 mmol/g of SCL.

The ESI value of the lipophilized protein had a similar trend even though, unlike EAI, it was significantly higher than that detected with the unmodified SCL at all levels of lipophilization. It is well known that the emulsifying activity depends on the formation both of a charged layer and of an elastic film around the oil droplet, determined by the proteins located at the interfaces that cause the repulsion of the droplets, whereas the strength of the protein-protein interactions at the oil/water interfaces determines the emulsion stability. Most probably SCL lipophilization exposed the hydrophilic groups of the protein and, at the same time, oriented its hydrophobic regions towards the lipid phase, thereby increasing the emulsifying properties of SCL [34]. However, the SCL emulsifying properties were observed to decrease at the higher level of lipophilization, probably because the hydrophilic groups available for the orientation towards the aqueous phase at the oil-water interface decreased excessively in the protein acylated with higher CAC amounts and, consequently, the formation of an elastic protein film at the interface might be hindered [25,35]. Similar results were reported by Akita and Nakai [9] in their studies on the lipophilization of β-lactoglobulin by stearic acid.

Finally, the effects of lipophilization on FC of SCL and FS at pH 7 are shown in Table 2.

It is well known that the FC depends on the ability of a protein to reduce the interfacial tension at the air-solution interface, which is usually accomplished by the unfolding and aligning of the proteins between the two phases [36]. Unmodified SCL was found to resist adsorption and unfolding at the air interface probably owing to its compact structure and, consequently, the formation of a suitable film was prevented. Conversely, the FC of SCL markedly increased by increasing its lipophilization with higher CAC amounts, clearly showing an improvement in the protein ability to trap air bubbles. Moreover, the FS, a measure of the ability of the foam to keep its maximum volume for a specific period [36], was also investigated for both unmodified and lipophilized SCL. The foams produced with lipophilized SCL were more stable than those obtained with unmodified SCL, so that ~61, 54, and 51% of the initial volume of foams obtained with SCL treated with 2, 4, and 6 mmol CAC, respectively, were still preserved after one-hour storage. These findings, together with those indicating an increased surface hydrophobicity and a reduced water solubility of the acylated SCL, suggest that protein-protein hydrophobic interactions are of crucial relevance for achieving stable films around air bubbles [36]. The decrease in the stability of the foams obtained with SCL lipophilized with higher CAC amounts (6 mmol/g), with respect to those obtained with SCL lipophilized with 2 mmol/g CAC, could be explained with a possible wide neutralization of the ε-amino groups of SCL lysine residues occurring following the acylation reaction. In fact, it is well known that the amino groups showed a higher tendency to participate in acylation reactions, due to the less steric hindrance, compared to other nucleophilic groups [25]. Therefore, the derived marked increase of the net negative charge of extensively acylated SCL might hinder the protein–protein interactions to form a continuous network around the air bubbles. In conclusion, the hydrophobic interactions would be predominant in SCL lipophilized with lower CAC amounts, while electrostatic repulsion would play a significant role in lowering the stability of foams obtained with SCL lipophilized at higher CAC concentrations.

Several acylation procedures carried out to modify the structure of different proteins of food interest have been previously developed in the attempt to improve their functional properties. Table 3 reports the most recent results to be compared with the present findings obtained with SCL. 

**Table 3 foods-10-00515-t003:** Effects of lipophilization of different proteins of food interest on their functional properties *.

Protein	Acylation Agent	Solubility	EAI	ESI	FC	FS	WAC	OAC	SHI
**Secalin**	CAC 2 mmol/g	**SD** (69%)	**SI** (56%)	**SI** (198%)	**SI** (167%)	**SI** (67%)	**SD** (33%)	(**SI** 83%)	**SI** (39%)
	CAC 4 mmol/g	**SD** (67%)	**SI** (12%)	**SI** (174%)	**SI** (270%)	**SI** (38%)	**SI** (50%)	**SI** (124%)	**SI** (78%)
	CAC 6 mmol/g	**SD** (66%)	**SD** (11%)	**SI** (106%)	**SI** (315%)	**SI** (30%)	**SD** (68%)	**SI** (136%)	**SI** (93%)
**Pea proteins [37]**	Succinic anhydride 0.03 g/g	**SI** (204%)	**NR**	**SI** (233%)	**SI** (64%)	**SI** (200%)	**SI** (107%)	**SD** (19%)	**NR**
**Wheat gluten proteins [38]**	Succinic anhydride 1.0 g/g	**SI** (583%)	**SI** (167%)	**SI** (112%)	**SI** (267%)	**SI** (457%)	**NSC**	**NR**	**NR**
**Mung bean proteins [31]**	Succinic anhydride 0.1 g/g	**NSC**	**SI** (42%)	**NSC**	**NR**	**NR**	**NR**	**NR**	**SD** (20%)
**Soy protein-7S [39]**	Caproic acid	**NR**	**SI** (50%)	**SI** (18%)	**NR**	**NR**	**SI** (192%)	**SI** (65%)	**SI** (47%)
	Caprylic acid	**NR**	**SI** (76%)	**SI** (27%)	**NR**	**NR**	**SI** (844%)	**SI** (35%)	**SI** (41%)
	Capric acid	**NR**	**SI** (86%)	**SI** (41%)	**NR**	**NR**	**SI** (760%)	**SI** (54%)	**SI** (168%)
	Lauric acid	**NR**	**SI** (100%)	**SI** (60%)	**NR**	**NR**	**SI** (388%)	**SI** (76%)	**SI** (154%)
	Myristic acid	**NR**	**SI** (70%)	**SI** (50%)	**NR**	**NR**	**SI** (745%)	**SI** (90%)	**SI** (138%)
	Palmitic acid	**NR**	**SI** (72%)	**SI** (27%)	**NR**	**NR**	**SI** (423%)	**SI** (44%)	**SI** (175%)
	Stearic acid (10.5 µmol/g)	**NR**	**SI** (70%)	**SI** (36%)	**NR**	**NR**	**NSC**	**SI** (118%)	**SI** (112%)
**α-lactalbumin [8]**	Lauroyl chloride	**SD** (31%)	**SI** (23%)	**SD** (62%)	**SI** (16%)	**SI** (24%)	**NR**	**NR**	**NR**
	Palmitoyl chloride	**SD** (3%)	**NSC**	**SD** (54%)	SD (32%)	**SI** (54%)	**NR**	**NR**	**NR**
	Stearoyl chloride (0.5 g/g)	**SD** (17%)	**NSC**	**SD** (35%)	**SD** (17%)	**SI** (63%)	**NR**	**NR**	**NR**
**Whey protein microgels [40]**	Acetic anhydride 650 mmol/mmol	**NR**	**NSC**	**SI** (15%)	**SI** (20%)	**SI** (46%)	**SD** (23%)	**SI** (32%)	**NR**
**Rapeseed proteins [41]**	Maleic anhydride 0.2 g/g	**SI** (27%)	**SI** (52%)	**SI** (57%)	**NSC**	**NSC**	**NR**	**NR**	**SI** (21%)

* All the data were obtained at pH 7; FS was determined after 30 min; **SI**, significantly increased; **SD**, significantly decreased; **NSC**, not significantly changed; **NR**, not reported. The values in parentheses show the differences compared to the control samples.

A water solubility decrease similar to that reported in the present study was obtained following lipophilization of α-lactalbumin with fatty acid chloride chains longer than CAC [8], even though SCL lipophilization by CAC was found to be more effective. The improvement in SCL emulsifying properties after lipophilization was comparable to the results of other studies so that EAI value observed with SCL acylated with low CAC amounts (2 mmol/g) was almost similar to those observed using both soy protein-7S acylated with caproic acid [39] and rapeseed proteins acylated with maleic anhydride [41]. Moreover, ESI values of lipophilized SCL were close to those reported for pea and wheat gluten proteins acylated using succinic anhydride [37,38] and higher than those reported for acylated soy protein-7S, whey protein microgels, and rapeseed proteins [39,40,41]. On the other hand, the foaming capacity improvement of lipophilized SCL was very impressive when compared to all other previous results, even though the stability of the produced foams was similar to that of α-lactalbumin lipophilized with either lauroyl, palmitoyl, or stearoyl chlorides [8] as well as to that of whey protein microgels acetylated by acetic anhydride [40]. Conversely, WAC values of lipophilized SCL, unlike the most reported acylated proteins, showed a decreasing trend, while the increase observed in OAC values was similar to that reported for almost all the other acylated proteins. Finally, Matemu et al. [39] showed that soy protein-7S acylation by medium-, and mostly long-chain, fatty acids was able to increase the SHI more than protein acylation by short-chain fatty acids, these results being similar to those obtained by acylating SCL with high amounts of CAC.

## 4. Conclusions

Changes in functional properties of SCL by capric acid incorporation were investigated. Increased protein surface hydrophobicity, oil absorption capacity, the emulsifying and foaming capacity of the lipophilized SCL, as well as the stability of both emulsions and foams obtained, were detected. Conversely, a marked decreased solubility of the protein at different pH values and at both low and high saline concentrations, as well as of its water absorption activity, were observed following SCL lipophilization. All the observed effects might be dependent on the protein unfolding and the formation of an elastic and stable protein film by hydrophobic interactions at the interfaces consequent to the capric acid covalent binding to SCL [9,20]. These findings strongly suggest the great potential of lipophilized SCL as an emulsifying and/or foaming additive in different food products.

## Figures and Tables

**Figure 1 foods-10-00515-f001:**
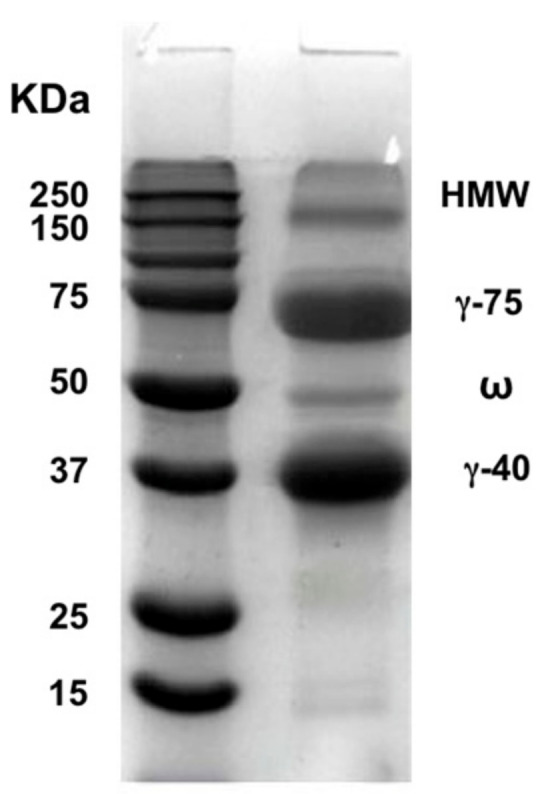
SDS-PAGE pattern of extracted secalin.

**Figure 2 foods-10-00515-f002:**
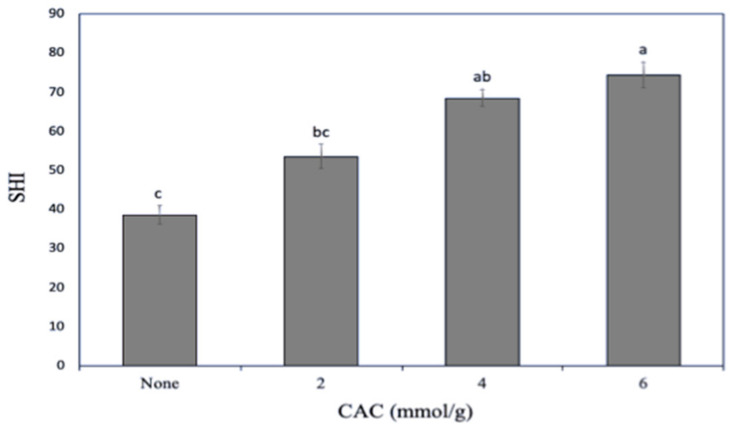
Effect of increasing capric acid chloride (CAC) amounts on the secalin surface hydrophobicity index (SHI). The lowercase letters (a–c) indicate significant differences among the values reported in each bar (*p* < 0.05). Further experimental details are given in the text.

**Figure 3 foods-10-00515-f003:**
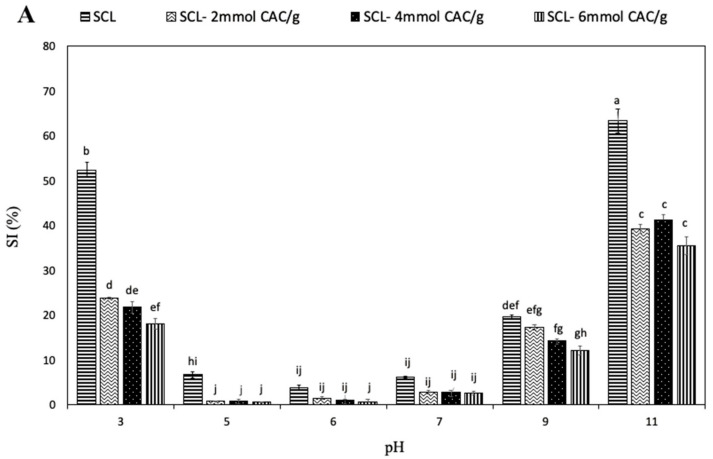
Effect of pH (**A**) and NaCl concentration (**B**) on the solubility index (SI) of secalin (SCL) treated or not with different amounts of capric acid chloride (CAC). The lowercase letters (a–j) indicate significant differences among the values reported in each bar (*p* < 0.05). Further experimental details are given in the text.

**Table 1 foods-10-00515-t001:** Effect of secalin (SCL) lipophilization with increasing capric acid chloride (CAC) on protein water (WAC) and oil absorption capacities (OAC) and on its emulsifying activity (EAI) and emulsion stability index (ESI) *.

SCL	WAC (mL/g)	OAC (mL/g)	EAI (m^2^/g)	ESI (min)
Unmodified	1.23 ± 0.04 ^a^	1.05 ± 0.07 ^c^	77.93 ± 1.82 ^c^	21.04 ± 0.64 ^c^
Acylated (2 mmol/g CAC)	0.83 ± 0.04 ^b^	1.92 ± 0.10 ^b^	121.41 ± 0.78 ^a^	62.66 ± 1.41 ^a^
Acylated (4 mmol/g CAC)	0.62 ± 0.03 ^c^	2.35 ± 0.06 ^a^	87.14 ± 2.34 ^b^	57.68 ± 1.55 ^a^
Acylated (6 mmol/g CAC)	0.39 ± 0.05 ^d^	2.48 ± 0.02 ^a^	69.27 ± 1.56 ^d^	43.37 ± 1.33 ^b^

* The lowercase letters (a–d) indicate significant differences among the values reported in each column (*p* < 0.05). Further experimental details are given in the text.

**Table 2 foods-10-00515-t002:** Effect of secalin (SCL) lipophilization with increasing capric acid chloride (CAC) on protein foaming capacity (FC) and foam stability (FS) *.

SCL	FC (%)	FS (%)		
		20 min	40 min	60 min
SCL	53.93 ± 3.23 ^d^	77.69 ± 3.81 ^a^	63.51 ± 1.08 ^ab^	39.33 ± 2.59 ^c^
Acylated (2 mmol CAC)	160.21 ± 2.82 ^c^	80.03 ± 4.06 ^a^	68.12 ± 0.56 ^a^	60.65 ± 2.46 ^a^
Acylated (4 mmol CAC)	199.53 ± 3.53 ^b^	76.19 ± 0.64 ^a^	64.18 ± 3.26 ^ab^	54.11 ± 2.58 ^b^
Acylated (6 mmol CAC)	248.45 ± 8.48 ^a^	69.77 ± 1.25 ^a^	60.28 ± 0.68 ^b^	50.98 ± 1.39 ^b^

* The lowercase letters (a–d) indicate significant differences among the values reported in each column (*p* < 0.05). Further experimental details are given in the text.

## Data Availability

Data available in a publicly accessible repository.

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
