# Peer review of "Functional Properties of Rye Prolamin (Secalin) and Their Improvement by Protein Lipophilization through Capric Acid Covalent Binding"

_foods, 2021, doi:10.3390/foods10030515_

Round 1
Reviewer 1 Report
general comments
This manuscript described modification of rye prolamin by covalent attachment of a fatty acid and its effect on foam and emulsifying properties. Tests were acceptably performed and standard. Writing was generally acceptable, although there were several sentences that used excessive number of clauses and were very difficult to read. Authors also had a tendency to utilize words inappropriate for the sentences, and authors may consider using simpler words whenever possible. Some conclusions were not sufficiently based on shown data, such as charge of proteins, interactions at the interface, elasticity of proteins at the interface, or conformation of the protein. Otherwise, the basic data is interesting and useful for researchers in the field. It would be very important to include IR spectra and carbon analysis data to prove covalent modification. Authors skip over this important section of the study very quickly.
Specific comments
lines 16-17: wording of 'emulsifying both capacity and stability were observed' is odd; would recommend revising to 'both emulsifying capacity and stability were improved' or similar.
lines 49-51: sentence requires revision
line 52: the 'gap in knowledge' statement is somewhat confusing, as it would be difficult to state whether addition of hydrophobic groups to the protein would improve properties if the initial properties are unknown. Rather, some general properties of secalin should be known; at the very least, it is a prolamin, which can say something of its relative solubility and (typical) functions.
line 53: ' the main...' should probably be a new sentence
line 62: was there any published detail on CAC or impurity content?
line 68: meaning of 'ratio of 1:10 (w/v)' is unclear. Does this mean 1 gram per 10 mL?
line 69: stating CAC concentration as mmol implies that the CAC was added as a solution (this may not be true, but this is the impression that it gave). Was this incoporated as a solution? If not, then it may be more straightforward to state v/v addition or w/w.
lines 71-72: it may be helpful to provide a tolerance to the acceptable final pH. Also, please note whether this was done manually with a benchtop meter or a titration unit.
line 80: scan rate and resolution for FTIR needed. Also, more details are needed on determination of lipophilization degree, as it is too vague to know how to replicate. This is in contrast with description of surface hydrophobicity, which is described in fine detail yet is a well known technique.
lines 139-144: authors describe lipophilization degree of only one of the treatments. Please provide data for other two treatments. Data on carbon content and IR spectra should be provided as supplementary data to support production of modified protein.
lines 196-199: sentence needs revision - too many clauses that are not well connected.
lines 200-201: second half of sentence was difficult to read. There is little relation between how ESI is similar to unmodified protein and is also greater. Wording needs to be changed.
line 201: change 'the ability to give rise to emulsions'. One example could be 'emulsifying activity'
lines 201-203: emulsion stability is not always dependent upon charged, elastic interfaces; this is only for certain systems. It would be best to modify this statement.
lines 209-210: please correct 'led to expose' and 'addressed the hyodrphobic regions to be oriented'; these are improper english phrases
lines 228-229: findings in this study did not demonstrate hydrophobic interactions between proteins at the interface; at most, authors could make a connection between increased surface hydrophobicity, decreased solubility in water, and increased foam stability.
lines 230-236: authors' logic is unclear, as data did not show an increased negative charge of proteins with increased CAC substitution. The discussion was then difficult to follow. Perhaps if the authors use electrophoresis to determine charge, then the text could be preserved. .
line 251: 'of soy protein...' is not properly connected with prior section of sentence. please revise.
table 3: according to the authors definition of solubility for this paper, maximum solubility would be 100%. references 36 and 37 showed >100% solubility. Please explain or revise.
Table 3: 'SI' was previously used for 'solubility index', so either this abbreviation or the previous should be changed to avoid confusion.
Reviewer 2 Report
Recommendation: accept
Comments to Authors: Zeinab Qazanfarzadeh, Mahdi Kadivar, Hajar Shekarchizadeh, and Raffaele Porta
Manuscript Number: foods-1051280-peer-review-v1
Article Type: Article
Article Title: Functional properties of rye prolamin (secalin) and their improvement by protein lipophilization through capric acid covalent binding
Overview and general recommendation
The introduction provides a sufficient background and contains relevant references to the problem raised.
The methods were presented correctly and chronologically adequate to the conducted research and are adequately described.
The research design is appropriate.
Overall Recommendation
In my opinion is accept

Reviewer 3 Report
In general, this study has been very well-organized and English is very clear. The results are giving an interesting and valuable information. However, there are some problems and flaws in presentation and discussion. I hope that my comments are very useful for the improvement of this research.
Comments
- Why did you choose capric acid chloride out of all the fatty acids? Please indicate the reason. It needs to be considered in comparison to other fatty acid in I Discussion.
- Statistical analysis: Authors used the Duncan’s multiple range tests as statistical analysis. But, Duncan's multiple range test has been pointed out to have problems such as not taking multiplicity. Thus, please change to another multiple tests.
- Discussion: Provide a discussion of the results of this experiment based on the molecular weight and amino acid composition of secalin. The current discussion is not based on the characteristics of secalin.
Reviewer 4 Report
In the manuscript entitled: “Functional properties of rye prolamin (secalin) and their improvement by protein lipophilization through capric acid covalent binding”, Qazanfarzadeh and co-authors lipophylized the secalin, a protein extract from rye prolamin, acylating it with capric acid chloride. Furthermore, the authors evaluated some functional properties such as the emulsifying and foaming properties of the lipophilized secalin.
Analytical comments:
Abstract
This section requires one or two general sentences to introduce the topic. This is important for non-expert readers. For example, what is secalin? Why is it important to lipophilize secalin?
Material and Methods
- The authors made some changes to the methods already described in the literature. However, these methods (i.e. those used to extract secalin and to evaluate functional properties of secalin) should be briefly explained.
Results
- It is not clear which proteins (proteins γ-75 KDa, ω and γ-40 KDa) the Authors used in their experiments. Please clarify and add an SDS-PAGE of the secalin extract.
- It is not clear why the Authors used capric acid chloride to acylate secalin. Please clarify.
- The Authors performed FTIR analysis. However, FTIR spectra are not present in this manuscript.
- In the figures, the authors indicated statistical significance using lowercase letters. However, the difference between the value indicated with the letter a and that with the letter b is not clear. In addition, in Figure 2 the authors used the letters a and b to indicate both the figure panel and the statistical significance, this make difficult to understand the figure and the caption. Letters e-m are not described in the Figure 2 caption.
- SCL solubility. The pI of SCL is 5.5, however the protein solubility in the pH range 5-7 is very similar. In addition, I have not found any experimental evidence to support that lipophilization decrease the SCL pI. How did the authors calculate / determine the pI of SCL and lipophilized SCL? The authors should explain why they tested the NaCl effect at pH 7. How do the Authors explain that lipophilization decrease the SCL solubility in all the tested conditions?
- The extrinsic fluorescent emission spectra should be shown. Indeed, the blue shift of the emission spectra is a hallmark of improved hydrophobicity.
- Results reported in table 3 should be better discusses and compared with those obtained in this paper.
Conclusion
- The conclusion: “All the observed effects might be dependent on the protein unfolding and the formation of an elastic and stable protein film by hydrophobic interactions at the interfaces consequent to the capric acid covalent binding to SCL” is not supported by experimental data. Concerning the protein unfolding, the Authors should be compared the FTIR spectra in the Amide I regions (1700–1600 cm−1) and fluorescence spectroscopy to observe changes in secondary and tertiary structure induced by lipophilization. See for example: https://doi.org/10.1016/j.ijbiomac.2020.02.145.
- It is not clear why secalin should be a good candidate to replace meat, egg and milk proteins. Please clarify. The potential of secalin to replace meat, milk, or egg proteins should be introduced in the Introduction section.
Minor comments:
Pag.1. Line 29: The Authors wrote: “the electrophoretic pattern of SCL shows four groups of polypeptides”. However, they reported 3 groups with different molecular weight, namely γ-75 KDa, ω and γ-40 KDa proteins. Please clarify and add the molecular weight of ω.
Pag.1 Line 33. It is not clear the relationship between β-sheet and protein amphiphilic behavior. Please clarify.
Pag.3 Line 95. What do the Authors mean with “and their pH adjusted? What value?
Pag 4. Line 136. What do the Authors mean with endoprotein amino acids?
Pag 4. Line 159. Why the solubility is an important property in food proteins?
Pag 6. Table 2. Table 2 should be moved in Pag 7.
Round 2
Reviewer 4 Report
Dear Editor,
In the revised manuscript, the authors have been responded properly to all of comments to add descriptions and make corrections. However, one point remains still open.
The Authors did not include the data concerning the characterization of acylated secalin, because these data have been reported in Qazanfarzadeh et al.. To reproduce the results described in this manuscript the FTIR spectra and elementary composition analysis should be reported at least as supplementary materials. If the authors used the SAME preparation of acylated secalin used in Qazanfarzadeh et al, they should write it in the Materials and Methods section and rewrite this paragraph.
Minor comments:
Pag. Line 185. Please add at least one reference after “because it can affect both emulsifying and foaming properties”.
Figure 1. SDS-PAGE should be upright.
